# Foot arch rigidity in walking: *In vivo* evidence for the contribution of metatarsophalangeal joint dorsiflexion

**Daniel J. Davis**[ID]*, **John H. Challis**

The Biomechanics Laboratory, The Pennsylvania State University, University Park, PA, United States of America

* djd426@psu.edu

**Data Availability Statement:** Data underlying the results presented in this study can be found at: https://figshare.com/account/home#/projects/139471.

## Abstract

Human foot rigidity is thought to provide a more effective lever with which to push against the ground. Tension of the plantar aponeurosis (PA) with increased metatarsophalangeal (MTP) joint dorsiflexion (i.e., the windlass mechanism) has been credited with providing some of this rigidity. However, there is growing debate on whether MTP joint dorsiflexion indeed increases arch rigidity. Further, the arch can be made more rigid independent of additional MTP joint dorsiflexion (e.g., when walking with added mass). The purpose of the present study was therefore to compare the influence of increased MTP joint dorsiflexion with the influence of added mass on the quasi-stiffness of the midtarsal joint in walking. Participants walked with a rounded wedge under their toes to increase MTP joint dorsiflexion in the toe-wedge condition, and wore a weighted vest with 15% of their body mass in the added mass condition. Plantar aponeurosis behavior, foot joint energetics, and midtarsal joint quasi-stiffness were compared between conditions to analyze the mechanisms and effects of arch rigidity differences. Midtarsal joint quasi-stiffness was increased in the toe-wedge and added mass conditions compared with the control condition (both $p < 0.001$). In the toe-wedge condition, the time-series profiles of MTP joint dorsiflexion and PA strain and force were increased throughout mid-stance ($p < 0.001$). When walking with added mass, the time-series profile of force in the PA did not increase compared with the control condition although quasi-stiffness did, supporting previous evidence that the rigidity of the foot can be actively modulated. Finally, more mechanical power was absorbed ($p = 0.006$) and negative work was performed ($p < 0.001$) by structures distal to the rearfoot in the toe-wedge condition, a condition which displayed increased midtarsal joint quasi-stiffness. This indicates that a more rigid foot may not necessarily transfer power to the ground more efficiently.

## Introduction

Human foot rigidity is thought to be a hallmark of the evolutionary divergence from our ape ancestors [1]. Rigid human feet could in theory provide a more effective lever with which to push against the ground, making bipedal gait more efficient [2, 3]. A recent interest in foot

**Funding:** The authors received no specific funding for this work.

**Competing interests:** The authors have declared that no competing interests exist.

and shoe biomechanics has seen research devoted to understanding the mechanisms of this rigidity as well as its influence on performance of day-to-day and athletic tasks [4, 5]. However, the mechanisms behind foot rigidity in gait are not settled, leading to a lack of consensus on, for example, the source of beneficial evolutionary adaptations, the most effective prostheses designs, and targeted therapies for foot disfunction.

A widely touted theory holds that when dorsiflexing the metatarsophalangeal (MTP) joint, the concurrent raising of the medial longitudinal arch (MLA), known as the windlass mechanism, results in a more rigid lever for more efficient transfer of energy between the ankle and the ground [2]. This is purportedly due to increased tension in the plantar aponeurosis (PA) due to MTP joint dorsiflexion, which has been observed in both *in vitro* and finite element analyses [6, 7] (Fig 1). *In vivo*, shear wave elastography has also indicated that measures of PA tissue stress increase as MTP joint dorsiflexion increases [8, 9]. Caravaggi et al. [10, 11] proposed that 'pre-tensioning' the PA due to increased MTP joint dorsiflexion prior to foot contact in walking could increase MLA rigidity and account for the decrease in medial-foot plantar pressures with faster walking speeds documented by Pataky et al. [12]. These findings have thus positioned MTP joint dorsiflexion, through the PA, as a logical means by which the human foot increases its rigidity during stance.

Recently, however, investigators have cast doubt on the effect of MTP joint dorsiflexion on MLA rigidity [5, 13]. Under vertical loads, the MLA elongated more and absorbed and dissipated more energy with the MTP joint dorsiflexed versus plantarflexed, refuting the hypothesis that MTP joint dorsiflexion increases MLA rigidity [5]. However, the uniaxial, quasi-static nature of this experiment may not have sufficiently captured the increased foot rigidity which potentially occurs in more dynamic movements like gait. Kern et al. [13], studied the rigidity of the midtarsal joint (a joint used to represent the MLA) when participants walked normally and with added mass, a perturbation which increases the forces and moments experienced by the foot's joints and could thereby alter the function of structures which cross these joints. Rigidity was quantified using sagittal plane midtarsal joint quasi-stiffness, which is the slope of the resultant joint moment to joint angular excursion line. This quasi-stiffness increased when walking with added mass due to an increase in resultant midtarsal joint moment with similar angular excursion between added mass and control conditions. However, MTP joint angle and range of motion did not differ in conditions with increased sagittal plane midtarsal joint quasi-stiffness, as might be expected should MTP joint dorsiflexion be solely responsible for increased midtarsal joint rigidity. From these data, it appears that the MLA can become more rigid independent of MTP joint dorsiflexion, but whether this dorsiflexion can itself also increase MLA rigidity in walking remains unclear.

The purpose of the present study is therefore to measure the influences of increased MTP joint dorsiflexion and added mass on the quasi-stiffness of the midtarsal joint in walking. Additionally, the influence of MTP joint dorsiflexion on the energy profiles of the foot and its joints will be examined. Should increasing the dorsiflexion of the MTP joint elicit an increase in rigidity as proposed by the windlass mechanism, it would be expected that this increase would occur due to a decrease in midtarsal joint angular excursion (i.e., MLA flattening) for a given moment. If the angular excursion of the midtarsal joint is indeed decreased with increased MTP joint dorsiflexion, it is anticipated that the midtarsal joint would produce less negative work during stance. Given previous work indicating that the midtarsal joint quasi-stiffness can be increased independent of increased MTP joint dorsiflexion, the present study will investigate whether increased MTP joint dorsiflexion and added mass during walking increase midtarsal joint quasi-stiffness through different mechanisms. MTP joint dorsiflexion would be expected to increase midtarsal joint quasi-stiffness by increasing the force in the PA, whereas it is anticipated that PA force will not be different if midtarsal joint quasi-stiffness is increased via a different mechanism.

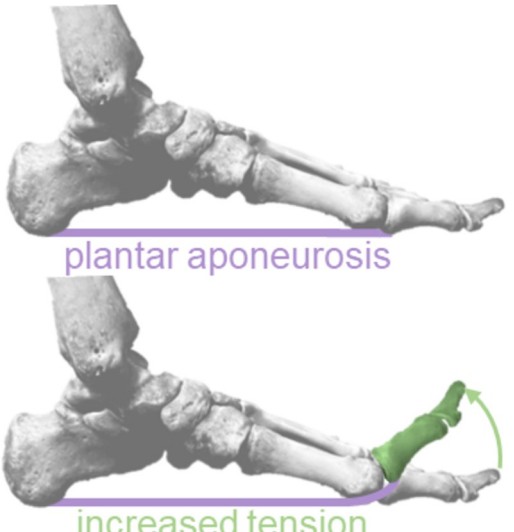

**Fig 1. Metatarsophalangeal joint dorsiflexion increases plantar aponeurosis tension.** The windlass mechanism, wherein dorsiflexion of the metatarsophalangeal joint increases the tension in the plantar aponeurosis, potentially increasing the rigidity of the foot's arch.

## Methodology

### Participants

Fourteen volunteers (12M/2F; 22 ± 4 yrs; 64.6 ± 11.7 kg; 1.71 ± 0.08 m) participated in the study. Potential participants were excluded if they had previous lower limb surgery, had been injured or had pain with walking or running in the past six weeks, or had ever had plantar fasciitis, diabetes, or osteoarthritis in their lower limbs. Participants gave oral informed consent to the study procedures which were approved by the Institutional Review Board of The Pennsylvania State University.

### Protocol

Target walking velocity was established using a Froude number of 0.22 [14]. Briefly, the Froude number scales gait velocity based on participant leg length (floor to participant's greater trochanter) and is used to produce 'dynamically similar' gait patterns for individuals with varying leg lengths [14]. The value of 0.22 was chosen as it would produce walking velocities typical in a healthy young population [15]. In all conditions, participants wore a minimalist sandal, which was constructed using a flexible running shoe insert with shoelaces to secure the participant's foot in the sandal. In the toe-wedge condition, a light-weight plastic wedge was adhered to the toe portion of the sandal (Fig 1). The bottom of this toe-wedge was rounded to dorsiflex the participant's MTP joints ~15 degrees while allowing participants to push off the ground naturally and minimally perturb the ground reaction force lever arm between conditions. The proximal edge of this wedge ran along the line between the first and fifth metatarsal heads. In the added mass condition, participants wore a weight vest which contained 15% of their body mass.

Participants walked across a Kistler force plate (Kistler Instrument Corp., Amhert, NY) at the prescribed velocities (± 5%) while marker positions were sampled at 200 Hz with eight Motion Analysis cameras (Motion Analysis Corp., Mountain View, CA) and ground reaction forces were sampled at 2,000 Hz. Participant starting position was adjusted to ensure their right foot landed entirely on the force plate. Participants were given as much time as necessary to familiarize themselves with walking in each condition and were instructed to walk as

naturally as possible. Three successful trials, defined as proper foot placement and walking velocity, were completed for each condition, and condition order was randomized.

## Foot model

The foot model used was similar to those of Caravaggi et al. [10] and Bruening et al. [16]. The model was made up of a shank, rearfoot, forefoot, and toe segments (Fig 2). Each foot segment was first defined using a long axis, then a geometric plane and a vector normal to the plane were defined. The remaining two segment axes were found using the cross products of the long axis and the vector normal to the plane. Bony landmarks for skin markers (diameter: 12.7 mm), joint centers, and segmental axes are depicted in Fig 2 and described in detail in S1 Appendix.

## Data analysis

Noisy marker position and ground reaction force signals were filtered at 6 Hz and 50 Hz, respectively, using a second order Butterworth filtered applied in forward and reverse

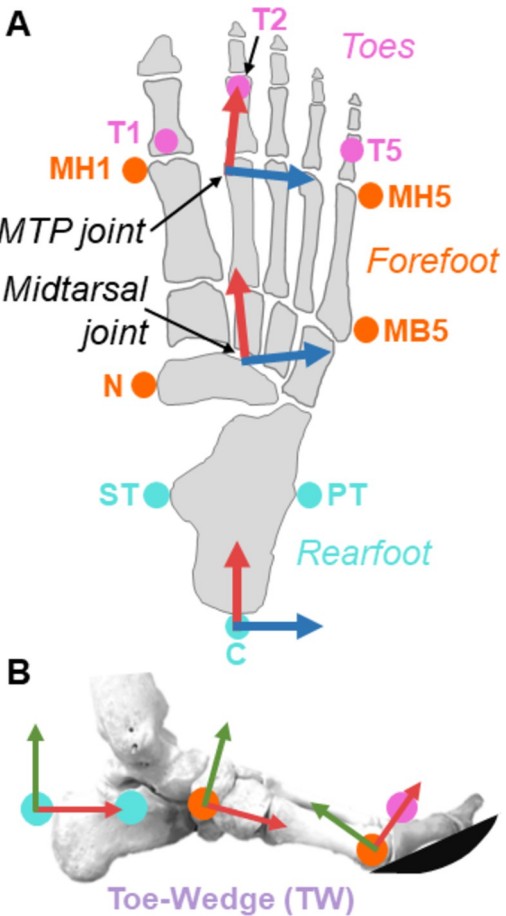

**Fig 2. Multi-segment foot model.** (A) Dorsal view of markers used to define the rearfoot (light blue; C = calcaneus, ST = sustentaculum tali, PT = peroneal tubercle), forefoot (orange; N = navicular, MH1 = metatarsal head 1, MH5 = metatarsal head 5, MB5 = metatarsal base 5), and toe (pink; T1, T2, and T5 = toe 1, toe 2, and toe 5, respectively) segments. Red arrows indicate the x-axis of the respective segment and blue arrows indicate the z-axis. (B) Medial view of markers defining foot segments. Red arrows indicate the segment's x-axis and green arrows indicate a segment's y-axis. The toe-wedge condition is depicted here wherein participants walked with a rounded toe-wedge which dorsiflexed their MTP joint.

directions. The position data cutoff frequency was selected after applying the autocorrelation-based procedure [17] on the stance phase marker trajectories. Foot strike and toe-off were defined as the time instance at which the vertical ground reaction force exceeded and fell below 35 N, respectively.

Three-dimensional Cardan joint angles were defined using a ZXY rotation sequence and expressed as distal to proximal segment rotations [18, 19]. For visualization, these rotations were expressed relative to the joint orientation in bilateral static stance, with positive values indicating dorsiflexion. Resultant joint moments were calculated using a Newton-Euler inverse dynamics approach and were expressed in the proximal segment reference frame, with positive values indicating an internal flexion moment. The midtarsal and MTP resultant joint moments were computed only after the center of pressure progressed anteriorly across each joint. MTP, midtarsal, and ankle joint six degrees of freedom power were calculated [20], as well as distal foot power which represents the power due to deformation of all structures distal to the estimated center of mass of the rearfoot segment (refer to [21] for distal foot power calculation details). Mechanical work was calculated as the time-integral of the power curves. Sagittal plane midtarsal joint quasi-stiffness was determined from the gradient of a straight line, which represented a least-squares fit to the non-dimensional sagittal plane resultant midtarsal joint moment versus joint angle data during the rise in the joint moment.

Resultant joint moments, power, and work are expressed as non-dimensional quantities. Resultant joint moments and work were normalized to the product of body mass, acceleration due to gravity, and leg length, and power was normalized to the product of body mass, acceleration due to gravity to the 3/2 power, and the square root of leg length [22]. In all cases, body mass not body mass plus added mass was used; normalizing in this manner quantifies how metrics changed with greater mass for a given individual, as opposed to normalizing to reduce differences due to different biological masses as is done between participants [23].

Estimated length of the PA was taken as the Euclidean distance from a virtual marker at the estimated PA attachment on the calcaneus to the MTP joint, plus the path length of the PA winding around the plantar aspect of the metatarsal heads (path length equals sagittal plane angle of the MTP joint multiplied by half the height of the MTP joint during static stance). The shortest length at which there was tension in the PA was estimated as the average of minimum PA lengths in each trial for each participant [24]. PA strain was calculated as the change in length relative to this shortest length at which there was tension in the PA. Force in the PA was estimated using the load-strain relationship of Wager [25] which was determined from tensile testing of five PA at 100 N/s [26]. The moment of the PA force about the midtarsal joint was taken as the cross product of the vector from the midtarsal joint to the PA and the force in the PA.

## Statistical analysis

Variables were averaged for each participant in each condition using the arithmetic mean or, for three-dimensional angles, using the singular value decomposition [27]. Effects of the toe-wedge and added mass on ankle and foot kinematics and kinetics were evaluated using discrete and time-series two-tailed paired t-tests in MATLAB (R2021b; Natick, Massachusetts) at $\alpha = 0.05$ using spm1d v0.4.8 [28]. A Bonferroni correction for the three comparisons (control versus toe-wedge, control versus added mass, and toe-wedge versus added mass) was applied for statistical inference, then p values were re-adjusted for reporting relative to 0.05. Normality was assessed using discrete and time-series forms of the D'Agostino-Pearson $K^2$ test [29]. For discrete variables, Bartlett's test for constant variance was used, whereas an F-test for equality of variance was used for time-series [30]. If either test was statistically significant, paired t-tests were conducted by generating probability density functions non-parametrically using 10,000

unique permutations of the experimental data [31]. A Bonferroni correction for the three comparisons (control versus toe-wedge, control versus added mass, and toe-wedge versus added mass) was applied for statistical inference, then p values were re-adjusted for reporting relative to $\alpha = 0.05$.

# Results

## Overall gait patterns

No statistically significant differences in average velocity or contact time were found for comparisons between the three conditions. Anterior-posterior ground reaction force profiles were slightly lower in magnitude in the toe-wedge condition than the control condition after mid-stance (p = 0.0018; S1A Fig), and greater in magnitude with the toe-wedge at the very end of stance (p = 0.0024; S1A Fig). There were no statistically significant differences in vertical or medio-lateral ground reaction force profiles or impulse in any plane between the toe-wedge and control conditions. Walking with added mass increased ground reaction force metrics compared with the control and toe-wedge conditions (see S1A–S1C Fig and S1 Table for details). The center of pressure excursion in mid- to late stance was further anterior between the toe-wedge and control conditions (p < 0.001), at 15% stance between the added mass and control conditions (p = 0.0475), and briefly in early (p = 0.011) and mid-stance (p = 0.0012) between the toe-wedge and added mass conditions (S1D Fig). The center of pressure excursion was more lateral at ~10% stance between the toe-wedge and added mass conditions (p = 0.02) (S1E Fig).

## Toe-wedge vs. control

Walking with the toe-wedge affected the kinetics and kinematics of the ankle and foot. In the toe-wedge condition, the MTP and ankle joints were significantly more dorsiflexed (both p < 0.001; Fig 3A and 3C). Participants also displayed greater sagittal plane MTP, midtarsal, and ankle resultant joint moment profiles in mid- to late stance when walking with the toe-wedge (all p < 0.001; Fig 3D–3F). The time-series profile of strain in the PA was greater in the toe-wedge condition than the control condition throughout mid-stance (p < 0.001; Fig 4) as was the profile of force in the PA (p < 0.001). Only the six degrees of freedom joint power profile at the MTP joint was altered in the toe-wedge condition compared with the control condition, with a more positive power profile in the toe-wedge condition (p = 0.0054; Fig 3G). The distal foot power profile was altered with the toe-wedge, with a greater negative power profile magnitude in mid-stance (p = 0.006; Fig 5A), and negative distal foot work was greater in magnitude in the toe-wedge condition (p < 0.001; Fig 5B and S2 Table). Finally, midtarsal joint sagittal plane quasi-stiffness was increased in the toe-wedge condition (p < 0.001; Fig 6B).

## Added mass vs. control

Walking with added mass also influenced joint angle profiles, resultant moment profiles, power profiles, work, and midtarsal joint quasi-stiffness. Non-dimensional MTP (p = 0.0048), midtarsal (p < 0.001), and ankle (p < 0.001) joint resultant moment profiles in the sagittal plane were greater in mid to late stance in the added mass condition (Fig 3D–3F). Midtarsal and ankle joint power profiles around the time of peak power were greater in the added mass condition (both p < 0.001; Fig 3H and 3I). Mechanical work was altered at each joint and in structures distal to the rearfoot center of mass, with greater magnitude of MTP joint negative work (p = 0.0036), greater midtarsal joint positive (p < 0.001) and magnitude of negative work

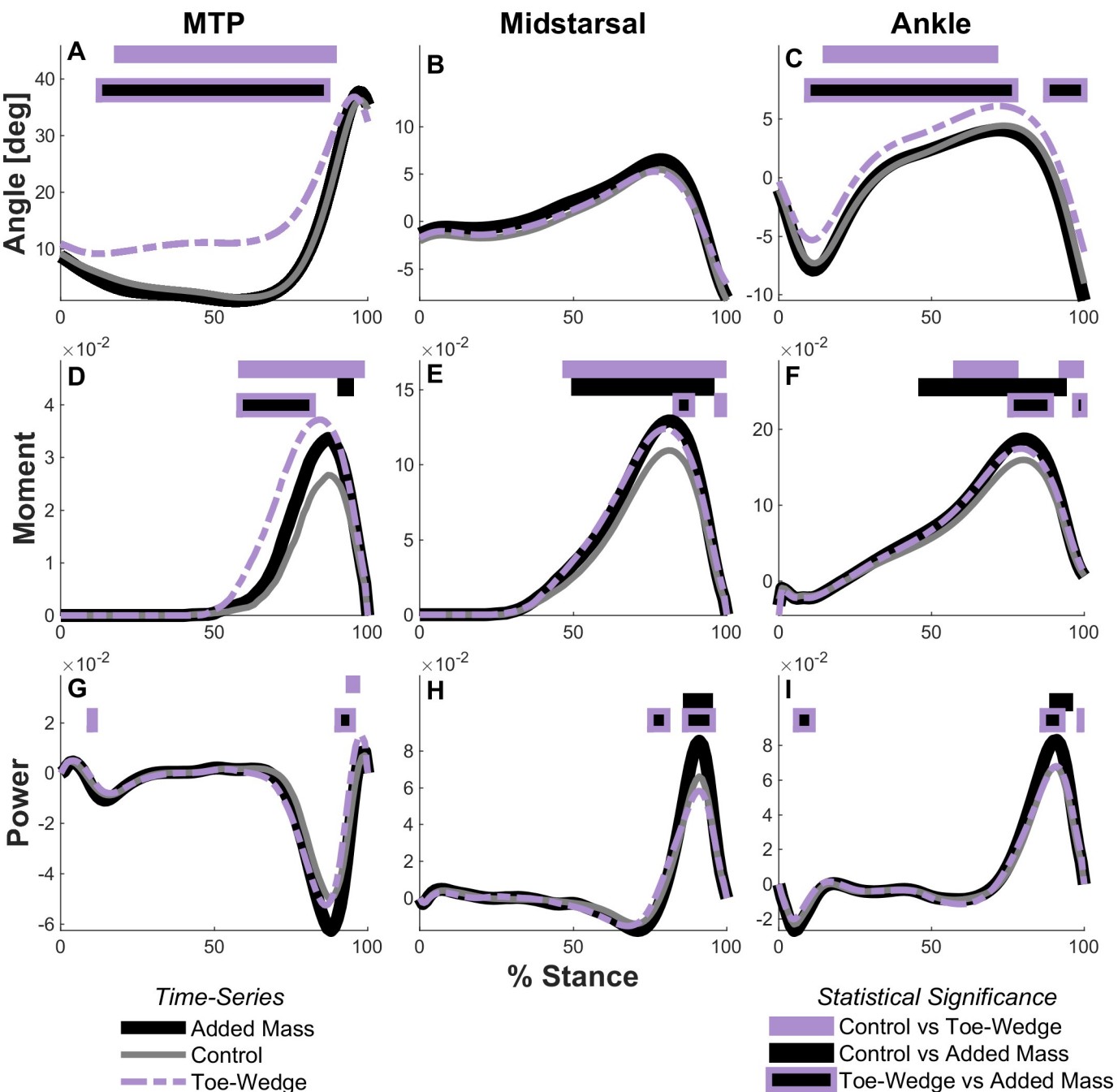

**Fig 3. Joint angles, resultant moments, and six degrees of freedom powers.** Time-series profiles of sagittal plane MTP, midtarsal, and ankle joint angles (panels A-C), resultant joint moments (panels D-F), and six degrees of freedom joint powers (panels G-I) in the added mass, control, and toe-wedge conditions. Moment and power are expressed as dimensionless quantities. Horizontal bars across the top of each panel indicate timings of statistically significant time-series differences between conditions using two-tailed paired t-tests ($\alpha = 0.017$ after Bonferroni correction).

($p = 0.0084$), greater positive ankle joint work ($p < 0.001$), and greater magnitude of distal foot negative work ($p = 0.0238$) when walking with added mass compared with the control condition (Fig 5B and S2 Table). Midtarsal joint quasi-stiffness increased from the control condition to the added mass condition ($p < 0.001$; Fig 6B).

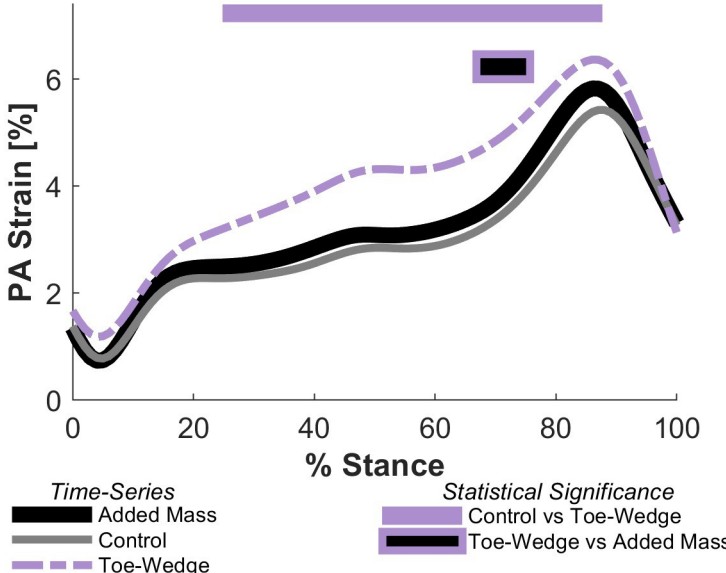

**Fig 4. Plantar aponeurosis (PA) strain.** Time-series profile of strain in the PA during stance for the added mass, control, and toe-wedge conditions. The shortest length at which there was tension in the PA for each participant was found using the mean of the shortest recorded PA lengths in each trial. The horizontal bars indicate timing of statistically significant time-series differences between the toe-wedge and control conditions (top) and between the toe-wedge and added mass conditions (bottom) using two-tailed paired t-tests ($\alpha = 0.017$ following Bonferroni correction).

### Toe-wedge vs. added mass

Walking with the toe-wedge elicited different gait kinematics and kinetics compared with walking with added mass. The MTP and ankle joint sagittal plane angle profiles were greater (more dorsiflexed) with the toe-wedge (MTP: $p < 0.001$; Ankle: $p = 0.012$ and $p < 0.001$, respectively; Fig **3A and 3C**). The resultant MTP joint moment profile in the sagittal plane was also greater with the toe-wedge ($p < 0.001$; Fig 3D). The resultant midtarsal and ankle joint moment profiles in the sagittal plane were greater with added mass in mid-stance (both $p = 0.0051$; Fig 3E and 3F) but were greater with the toe-wedge in late stance ($p = 0.0051$ and 0.0015, respectively; Fig 3E and 3F). In the toe-wedge condition, there were increased time-series profiles of strain and force in the PA compared with the added mass condition (both $p < 0.001$; Fig 4). Compared with the toe-wedge condition, the added mass condition elicited greater magnitudes of negative joint power profile at the MTP joint ($p = 0.0209$ and $p = 0.006$, Fig 3G), greater positive power profiles at the midtarsal joint (both $p < 0.001$, Fig 3H), greater magnitudes of negative ankle joint power profiles early in stance ($p = 0.0015$, Fig 3I), greater positive ankle joint power profile around the time of peak power ($p = 0.0015$, Fig 3I), and decreased ankle joint power profile at the very end of stance ($p = 0.0209$; Fig 3I). Lastly, the added mass condition displayed a greater magnitude of MTP joint negative work ($p = 0.022$), greater midtarsal joint positive work ($p < 0.001$), and greater ankle joint positive work ($p < 0.001$) compared with the toe-wedge condition (S2 Table).

### Discussion

The aim of the present study was to examine how increased MTP joint dorsiflexion and added mass would alter midtarsal joint quasi-stiffness in walking. Midtarsal joint quasi-stiffness was increased both when participants walked with a toe-wedge that increased MTP joint

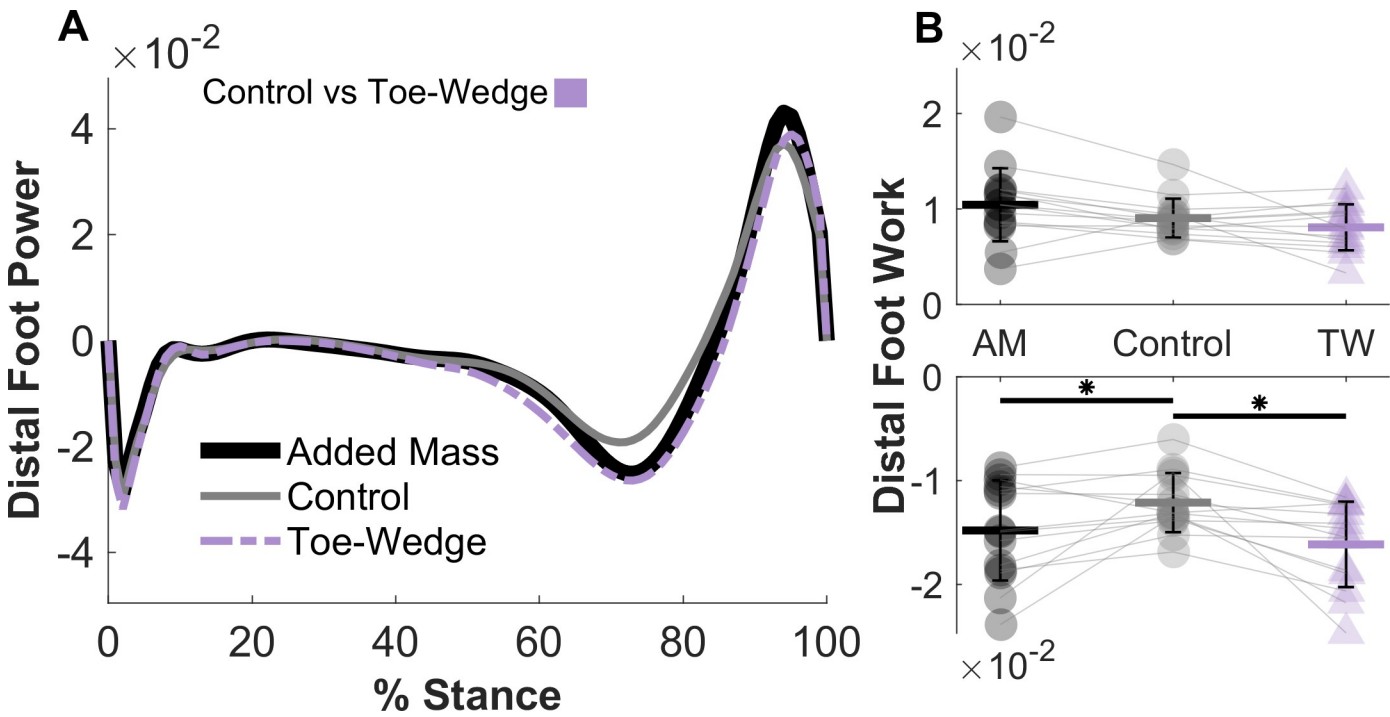

**Fig 5. Distal foot mechanical work and power.** (A) Mean mechanical power profiles and work (panel B) of structures distal to the rearfoot center of mass in the added mass, control, and toe-wedge conditions. In panel A, the horizontal bar indicates the timing of a statistically significant time-series differences between the control and toe-wedge conditions using a two-tailed paired t-test ($\alpha = 0.017$ after Bonferroni correction). (B) Distal foot negative work was greater in magnitude in the toe-wedge (TW) and added mass (AM) conditions than the control condition (* in panel B indicates statistical significance at $\alpha = 0.017$ after Bonferroni correction). Values for each participant are displayed in each condition, with grey lines connecting a single participant across conditions. Error bars represent the standard deviation.

dorsiflexion and with a weight vest that increased mass. The toe-wedge condition resulted in greater time-series profiles of strain and force in the PA, providing a mechanism for increased midtarsal joint quasi-stiffness. However, in contrast to the toe-wedge condition, PA strain and force profiles were not statistically significantly greater in the added mass condition compared with the control condition. These results, along with the results of Kern et al. [13], provide evidence that the increase in quasi-stiffness when walking with added mass is independent of increased PA tension. This study additionally aimed to assess the influence of increased MTP joint dorsiflexion on foot energetics in walking. Increased MTP joint dorsiflexion altered the mechanical power profile and work of structures distal to the estimated rearfoot center of mass, with more negative work in the toe-wedge condition compared with the control condition.

Estimates of the force in the PA and the moment this force produces about the midtarsal joint support the hypothesis that increased MTP joint dorsiflexion can increase the rigidity of the MLA. Between the toe-wedge and control conditions, the mean difference in PA moment about the midtarsal joint during the rise in resultant midtarsal joint moment was approximately 60% of the mean difference in resultant midtarsal joint moment, which suggests that other structures–either active or passive–are also responsible for the increase in resultant joint moment profile. This finding is supported by cadaver studies which indicate that removal of the PA does not remove all the MLA's rigidity [32, 33]. The increased time-series profile of strain in the PA in the toe-wedge condition is likely mirrored in tissues with similar paths like the long plantar ligament and the tendons of the flexor digitorum/hallucis longus and brevis [33], resulting in increased forces in these tissues as well.

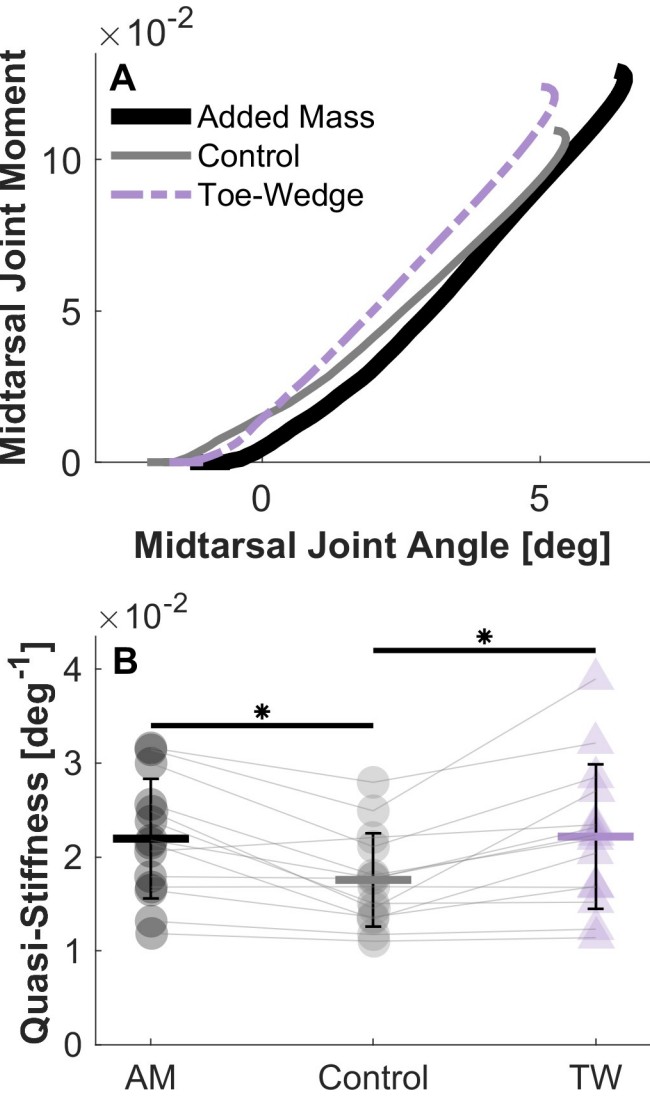

**Fig 6. Midtarsal joint sagittal plane quasi-stiffness.** (A) Mean sagittal plane resultant midtarsal joint moment profile and sagittal plane angle from the time the center of pressure crossed the joint to the maximum moment in the added mass, control, and toe-wedge conditions. (B) Sagittal plane midtarsal joint quasi-stiffness was greater in the toe-wedge (TW) and added mass (AM) conditions than in the control condition (* indicates statistical significance at $\alpha = 0.017$ after Bonferroni correction). Values for each participant are displayed in each condition, with grey lines connecting a single participant across conditions. Error bars represent the standard deviation.

The model of the PA employed here is geometrically similar to those in Caravaggi et al. [10] and Wager & Challis [24] and resulted in mean peak PA strain of approximately 5% in the control condition, which agrees well with *in vivo* [10] and *in vitro* [34] findings. Mean peak forces in the PA were near one body weight in the control condition (0.95 body weights), which is consistent with *in vitro* and finite element measures [20, 35], but lower than those estimated by Giddings et al. [36] (~1.8 body weights) and Caravaggi et al. [10] (~1.5 body weights). If PA forces were underestimated in the present study, accounting for this potential discrepancy would provide further support that MTP joint dorsiflexion can bolster MLA rigidity.

Neither the time-series profiles of PA strain, force, or moment about the midtarsal joint were statistically significantly different when comparing the added mass and control

conditions. Similar to the conclusions from others [11, 13], this supplies additional evidence that active mechanisms, not the functioning of the passive PA, are likely responsible for the increase in midtarsal joint quasi-stiffness when walking with added mass. Farris et al. [37] indicated that blocking the activity of the intrinsic foot muscles did not change the quasi-stiffness of the MLA, but this does not mean that the intrinsic foot muscles are not capable of increasing this quasi-stiffness. Indeed, Kelly et al. [38] showed that electrostimulation of these muscles can increase the MLA's rigidity during vertical loading. Further, previous studies have demonstrated that the foot muscles can increase the rigidity of the MTP joint in walking [37, 39]. Echoing these results, in the present study the resultant MTP joint moment profile was increased, and the joint angle profile was not, therefore the quasi-stiffness of the MTP joint would indeed have increased in the added mass compared with the control condition. While it cannot be confirmed in the present study, the results align with growing evidence that the muscles of the foot are capable of modulating its rigidity [38–40].

The finding that midtarsal joint quasi-stiffness was increased with increased MTP joint dorsiflexion is contradictory to Welte et al. [5], who did not find an increase in measures of MLA rigidity during vertical loading with MTP joint dorsiflexion. This is likely in part due to differences between the foot's behavior in vertical loading and gait. For example, Farris et al. [37] found that blocking the activity of intrinsic foot muscles decreased MLA rigidity in vertical loading, but not in gait. Further, Welte et al. [5] did not quantify midtarsal joint quasi-stiffness, but instead measured MLA angular excursion, MLA compression and elongation, and navicular tuberosity motion. Midtarsal joint quasi-stiffness may have in fact increased in that study given that MLA elongation increased, and MLA angular excursion did not. If the MLA is lengthened, the center of pressure under the forefoot would likely be further from the midtarsal joint, generating a greater moment for a given force. A greater moment without a concomitantly greater angular excursion could result in greater quasi-stiffness. The discrepancy between the results of the present study and those of Welte et al. [5] underly the influence of the type of loading on the foot's response, as well as the importance of consistent methods for quantifying the rigidity of the MLA between studies.

It is perhaps paradoxical that a foot with a more rigid midtarsal joint would dissipate more energy, as was seen when participants walked with the toe-wedge. The distal to rearfoot power profile and work incorporate the functioning of the midtarsal and MTP joints and were both greater in absolute magnitude in the toe-wedge condition compared with the control condition during a phase of stance in which both joints were also absorbing power (Figs 3G and 3H and 5). The finding that work from structures distal to the rearfoot was altered with increased MTP joint dorsiflexion aligns with Welte et al. [5] in that the foot absorbed more energy when vertically loaded with the MTP joint dorsiflexed. Riddick et al. [41] also found the midtarsal joint performed more negative work during hopping with an exotendon that increased measures of foot rigidity. The extent of potential benefits from MLA rigidity must be tempered by the fact that the MTP joint also plays a substantial role in power transfer to the ground, with peak power absorption magnitudes of more than half the peak power generation at the ankle joint (Fig 3G). A more rigid MTP joint, which was seen here in toe-wedge condition judging by the increased resultant joint moment without greater angular excursion, can in turn lengthen the lever arm from the center of pressure to the midtarsal and ankle joints during push-off [42]. This longer lever arm could subsequently increase power absorption at the midtarsal joint due to greater moment and would also necessitate greater ankle plantarflexor muscle force, both of which may increase the metabolic power necessary to walk [42]. There is possibly a combination of midtarsal and MTP joint rigidity values that would best balance the power generation and absorption demands of gait, which is an investigation well-suited for computational models.

Limitations in the design and analysis of the present study provide context for the findings and conclusions. First, differences in muscular activation across conditions were not assessed, and therefore the extent to which differences in midtarsal joint quasi-stiffness are due to active versus passive mechanisms can only be approximated. Walking with the toe-wedge may have resulted in participants altering their overall gait patterns, although general metrics characterizing gait patterns–peak anteroposterior and vertical ground reaction force, contact time, and average velocity–were not statistically significantly different between toe-wedge and control conditions. The midtarsal joint quasi-stiffness increase in the toe-wedge condition compared with the control condition could be in part due to the increased ankle joint dorsiflexion angle profile, given the association between ankle and MLA angles [9, 43]. However, the ankle joint dorsiflexion profile was greater in the toe-wedge condition compared with the added mass condition as well, but there were no statistically significant differences in midtarsal joint quasi-stiffness between these conditions as might be expected should ankle joint dorsiflexion have caused the increase in MLA rigidity. The increase in ankle joint dorsiflexion profile in the toe-wedge condition can be attributed to the toe-wedge slightly raising the metatarsal heads and thus the remainder of the foot, not the participants altering their overall gait mechanics. This is evidenced by a lack of time-series difference in the dorsiflexion angle profile when angles are expressed relative to a static trial in which participants stood naturally with the toe-wedge, as opposed to the control static trial as is presented in Fig 3C. Nonetheless, a potential link between MLA rigidity and ankle joint orientation would be a fruitful avenue of future study in elucidating an additional mechanism for human foot rigidity regulation.

## Conclusions

The contribution of MTP joint dorsiflexion to MLA rigidity cannot be ruled out, given that when participants walked with increased MTP joint dorsiflexion there was more force in the PA which contributed to a greater moment profile about the midtarsal joint. The time-series profile of force in the PA, however, does not explain the increase in midtarsal joint quasi-stiffness when participants walked with added mass, providing further evidence that the rigidity of the foot can be actively modulated depending on task demands. Finally, more mechanical energy was absorbed by the foot when participants walked with increased MTP joint dorsiflexion, a perturbation that increased the foot's rigidity, providing nuance to the claim that increased foot rigidity is beneficial for energetically efficient human walking. Altogether, the present study experimentally demonstrates that foot arch rigidity can be increased via passive mechanisms (i.e., increased PA strain through MTP joint dorsiflexion) and supports previous evidence that active mechanisms are capable of increasing this rigidity as well. These findings should be considered when investigating the evolution of the human foot, interpreting its response to task demands, and when designing footwear and biomimetic prostheses.

## Supporting information

**S1 Appendix. Marker and joint names and locations, foot segment axes definitions, and foot segment descriptions.**
(DOCX)

**S1 Fig. Ground reaction force and center of pressure time-series profiles.** Panels A-C: Anterior-posterior (ap), vertical (v), and medial-lateral (ml) ground reaction force profiles (GRFs) normalized to body mass in all three planes in the added mass (AM), control, and toe-wedge (TW) conditions. Panels D-E: Anterior-posterior and medial-lateral center of pressure (CoP) profiles normalized to foot length in the added mass, control, and toe-wedge conditions.

Horizontal bars top of the left panel indicate the timing of a statistically significant time-series differences between the control and toe-wedge conditions using a two-tailed paired t-test ($\alpha = 0.017$ after Bonferroni correction). The top (purple) horizontal bar indicates a difference between control and toe-wedge conditions, the middle (black) bar represents a difference between the control and added mass conditions, and the bottom (black with purple outline) bar denotes a difference between the toe-wedge and added mass conditions.
(TIF)

**S1 Table. Impulse values determined from ground reaction force-time profiles.** Values represent mean ± standard deviation and are normalized to total mass in each condition.
(DOCX)

**S2 Table. Dimensionless joint mechanical work.** Values represent mean ± standard deviation and are normalized to the product of body mass, acceleration due to gravity, and leg length.
(DOCX)

## Author Contributions

**Conceptualization:** Daniel J. Davis.

**Data curation:** Daniel J. Davis.

**Formal analysis:** Daniel J. Davis.

**Methodology:** Daniel J. Davis, John H. Challis.

**Resources:** John H. Challis.

**Supervision:** John H. Challis.

**Visualization:** Daniel J. Davis.

**Writing – original draft:** Daniel J. Davis, John H. Challis.

**Writing – review & editing:** Daniel J. Davis, John H. Challis.

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
