## [Decision Letter · Decision Letter 0]

12 Aug 2022

PONE-D-22-15548Foot arch rigidity in walking: In vivo evidence for the contribution of metatarsophalangeal joint dorsiflexionPLOS ONE

Dear Dr. Davis,

Thank you for submitting your manuscript to PLOS ONE. After careful consideration, we feel that it has merit but does not fully meet PLOS ONE’s publication criteria as it currently stands. Therefore, we invite you to submit a revised version of the manuscript that addresses the points raised during the review process.

We look forward to receiving your revised manuscript.

Kind regards,

Imre Cikajlo, Ph.D.

Academic Editor

PLOS ONE

Journal Requirements:

Reviewers' comments:

Reviewer's Responses to Questions

**Comments to the Author**

1. Is the manuscript technically sound, and do the data support the conclusions?

Reviewer #1: Yes

Reviewer #2: Yes

2. Has the statistical analysis been performed appropriately and rigorously? 

Reviewer #1: No

Reviewer #2: Yes

3. Have the authors made all data underlying the findings in their manuscript fully available?

Reviewer #1: Yes

Reviewer #2: Yes

4. Is the manuscript presented in an intelligible fashion and written in standard English?

Reviewer #1: Yes

Reviewer #2: Yes

5. Review Comments to the Author

Reviewer #1: I have completed the evaluation of the manuscript entitled ‘Foot arch rigidity in walking: In vivo evidence for the contribution of metatarsophalangeal joint dorsiflexion’. Overall, this study is an interesting study, and I am satisfied with the manuscript quality. It investigated the gait biomechanical characteristics under the added mass and increasing MTP joint dorsiflexion conditions. It may have a potential use in a clinical setting for understanding the foot rigidity mechanism. However, I have some reservations that I hope the author(s) will address in the manuscript.

1. Abstract, the background is sufficient but could be tweaked a little bit to concretize the abstract. Furthermore, enhancing the methods in the abstract. The results should be focalized on the main findings.

2. Lines 52-54, providing the literature evidence to support your opinion.

3. The introduction is good but can be improved by illustrating different methods utilised in understanding foot windlass mechanisms and recent advances.

4. More detailed participant info, such as height, would be expected.

5. If three trials were sufficient to reduce the data collection errors?

6. Has the author (s) considered the analysis of variance test to check the statistical inference between groups. Please note that t-tests are NOT the non-parametric tests. The description of the statistical analysis section should be optimized.

7. I can not find the figure in the S1 appendix.

8. What is PA stand for in the caption of Fig 3? The readers would appreciate the full name and abbreviation in the figure.

9. I recommend splitting Fig 5 with panels A and B for moment and quasi-stiffness measures.

10. It’s a bit hard to understand why the black line (AM) is wider than others. Presenting data with Mean±Standard deviation may help to illustrate the data trend.

11. It is suggested that the figure legends be depicted below the subfigure Power and list in line.

Reviewer #2: Thank you for the opportunity to review this interesting paper. This study sought to explore the role of the windlass mechanism and medial arch rigidity on energy transfer during walking by increasing dorsiflexion at the metatarsophalangeal joint (MTPJ) angle and increasing body mass. The work is an extension of work by Welte et al. (2018) who manipulated MTPJ angle during vertical loading and the work of Kern et al. (2019) who explored the effect of increased mass on midtarsal quasi-stiffness in walking. The study would appear to be novel and well designed and delivered. My comments are largely around adding greater explanation of rationale, mechanisms and variables for those who are not so familiar with potentially challenging concepts.

Page 3, lines 58 to 61: Figure 1 in the paper by Welte et al. assists the explanation of the windlass mechanism well. The authors of the present study may wish to consider adding a similar figure.

Similarly, clarity could be improved by defining key variables (e.g. quasi-stiffness) and establishing their relevance early on. For example, in the introduction in the Kern et al. paper it is stated: “Quasi-stiffness of the ankle (sometimes called dynamic stiffness) is defined as the slope of the joints’ moment-angle relationship (Sanchis-Sales et al., 2016; Shamaei, Sawicki & Dollar, 2013; Rouse et al., 2013). This is an experimentally derived parameter, which describes the joints’ resistance to motion for a given change in moment throughout stance.”

Page 4, line 72: I would replace “In Kern et al” with “In a study by Kern et al” and “participant’s” with “participants’”.

Page 4, lines 72 to 80: The reporting of the findings of Kern et al is good, however I think the rationale for why increasing mass is worth investigating could be made clearer, perhaps with reference to altered forces experienced by the foot as the original authors did.

Page 5, line 98: Was there a justification for recruiting 14 participants? Was a power calculation performed?

Page 5, line 103: The flow of the Methodology (Materials and Methods according to PLOS One submission guidelines) may be improved by reporting the protocol before the foot model, as was the case in the earlier work by Welte et al. and Kern et al.

Page 5, line 101: According to PLOS One submission guidelines it should be specified whether informed consent was written or oral.

Page 5, line 108: What was the diameter of the markers?

Page 6, lines 118 to 126: For completeness I would add a specific definition of strain (length change relative to resting length?) in outlining how length of the PA was calculated. I would also consider placing this section in the data analysis along with the definition of the kinetic variables of interest.

Page 6, lines 130 to 132: Were the markers attached directly to the skin and not on socks?

Page 6, lines 128 to 129: The concept of a Froude number was interesting and not something I have come across before in gait or foot and ankle literature (and I see the reference is rather old). Is it known how this number differs from a self-selected speed? Could it be a point for discussion, seeing as Welte et al. compared loading at different speeds, although the only significant difference between speeds which was found as in energy dissipation? In any case I would recommend rearranging the sentence to start with something along the lines of “Target walking velocity was established by…” so the Froude number is not emphasized as much and does not detract from the main purpose.

Page 6, line 133: Is the shore value (hardness) of the wedge known?

Page 6, line 138: The previous work used 15% and 30%, why was a value of 15% chosen here?

Page 7, line 152: Was there a rationale for a threshold of 35 N, which is higher than I would expect?

Page 7, line 154: Is the rotation sequence correct? In reference 19 it is stated: “A ZYX Tait–Bryan angle sequence determined the angles of the first metatarsal relative to the calcaneus (arch angles) and the phalanx relative to the metatarsal (MTPJ angle)”.

Page 8, lines 163 to 164: How was power calculated (vertical GRF multiplied by arch velocity?).

Page 9, line 196 and subsequent references: I do not see a supplementary figure, only tables and text…

Page 10 to page 13: When referring to differences shown in Figure 2, readability could be improved by referencing the specific panel in Figure 2.

Page 11, lines 232 to 240: My interpretation is that the circles and triangles represent individual participants, if this is correct it may help to clarify in the figure caption? I would advocate such an approach as there is a lot of inter-individual variability in foot function, so it is useful to demonstrate whether differences in conditions was consistent across participants in addition to any difference in the means.

Page 16, lines 361 to 362: Was foot type/posture accounted for? How might this effect finings?

Page 16, lines 363 to 364: A “to” is missing from “[differences…] due active”

Figures 2-5: Clarity may be improved by writing added mass and toe wedge in full rather than using uncommon abbreviations.

6. PLOS authors have the option to publish the peer review history of their article (what does this mean?). If published, this will include your full peer review and any attached files.

Reviewer #1: **Yes: **Liangliang Xiang

Reviewer #2: **Yes: **Dr Joanna Reeves

---

## [Author Response · Author response to Decision Letter 0]

20 Aug 2022

Responses to Comments of Reviewer 1

Comment: I have completed the evaluation of the manuscript entitled ‘Foot arch rigidity in walking: In vivo evidence for the contribution of metatarsophalangeal joint dorsiflexion’. Overall, this study is an interesting study, and I am satisfied with the manuscript quality. It investigated the gait biomechanical characteristics under the added mass and increasing MTP joint dorsiflexion conditions. It may have a potential use in a clinical setting for understanding the foot rigidity mechanism. However, I have some reservations that I hope the author(s) will address in the manuscript.

Response: Thank you for your positive comments.

Comment: Abstract, the background is sufficient but could be tweaked a little bit to concretize the abstract. 

Response: A sentence was added to the abstract to better present pertinent background information. The background section of the abstract now reads,

 “Human foot rigidity is thought to provide a more effective lever with which to push against the ground. Tension of the plantar aponeurosis (PA) with increased metatarsophalangeal (MTP) joint dorsiflexion (i.e., the windlass mechanism) has been credited with providing some of this rigidity. However, there is growing debate on whether MTP joint dorsiflexion does indeed increase arch rigidity. Further, the arch can be made more rigid independent of additional MTP joint dorsiflexion (e.g., when walking walk with added mass).”

Comment: Furthermore, enhancing the methods in the abstract.

Response: The section of the abstract which pertains to the study methodology now reads,

 “Participants walked with a rounded wedge under their toes to increase MTP joint dorsiflexion in the toe-wedge condition, and wore a weighted vest with 15% of their body mass in the added mass condition. Plantar aponeurosis behavior, foot joint energetics, and midtarsal joint quasi-stiffness were compared between conditions to analyze the mechanisms and effects of arch rigidity differences.”

Comment: The results should be focalized on the main findings.

Response: A sentence has been removed so as to limit the presentation of results to those on the main findings. This sentence originally read,

 “There was also a negative power profile which was greater in magnitude in structures distal to the rearfoot in mid-stance (p = 0.006), and negative distal to rearfoot work was greater in magnitude in the toe-wedge condition compared with the control condition (p < 0.001).”

 These results were instead more succinctly mentioned in context of the studies’ conclusions in the closing of the abstract, which now reads,

 “Finally, more mechanical power was absorbed (p = 0.006), and negative work performed (p < 0.001), by structures distal to the rearfoot in the toe-wedge condition, a condition which displayed increased midtarsal joint quasi-stiffness. This indicates that a more rigid foot may not necessarily transfer power to the ground more efficiently.”

Comment: Lines 52-54, providing the literature evidence to support your opinion.

Response: A citation to the below articles has been included to support the statement in question:

Kelly LA, Lichtwark G, Cresswell AG. Active regulation of longitudinal arch compression and recoil during walking and running. J R Soc Interface. 2015;12(102): 20141076–20141076. doi:10.1098/rsif.2014.1076.

Welte L, Kelly LA, Lichtwark GA, Rainbow MJ. Influence of the windlass mechanism on arch-spring mechanics during dynamic foot arch deformation. J R Soc Interface. 2018;15(145): 20180270. doi:10.1098/rsif.2018.0270.

Comment: The introduction is good but can be improved by illustrating different methods utilised in understanding foot windlass mechanisms and recent advances.

Response: Some further details have been included in the second paragraph of the introduction to address methods utilized in understanding the windlass mechanism and recent advances. The sentences in question now read,

 “This is purportedly due to increased tension in the plantar aponeurosis (PA) due to MTP joint dorsiflexion, which has been observed in both in vitro and finite element analyses [6,7] (Fig 1). In vivo, shear wave elastography has also indicated that measures of PA tissue stress increase as MTP joint dorsiflexion increases [8,9]. Caravaggi et al. [10,11] proposed that ‘pre-tensioning’ the PA due to increased MTP joint dorsiflexion prior to foot contact in walking could increase MLA rigidity and account for the decrease in medial-foot plantar pressures with faster walking speeds documented by Pataky et al. [12].”

6. Carlson RE, Fleming LL, Hutton WC. The biomechanical relationship between the tendoachilles, plantar fascia and metatarsophalangeal joint dorsiflexion angle. Foot Ankle Int. 2000;21(1): 18–25. doi:10.1177/107110070002100104.

7. Cheng H-YK, Lin C-L, Chou S-W, Wang H-W. Nonlinear finite element analysis of the plantar fascia due to the windlass mechanism. Foot Ankle Int. 2008;29(8): 845–851. doi:10.3113/FAI.2008.0845.

8. Chino K, Lacourpaille L, Sasahara J, Suzuki Y, Hug F. Effect of toe dorsiflexion on the regional distribution of plantar fascia shear wave velocity. Clin. Biomech. 2019;61(2019): 11–15. doi:10.1016/j.clinbiomech.2018.11.003.

9. Shiotani H, Maruyama N, Kurumisawa K, Yamagishi T, Kawakami Y. Human plantar fascial dimensions and shear wave velocity change in vivo as a function of ankle and metatarsophalangeal joint positions. J Appl Physiol. 2021;130(2): 390–399. doi:10.1152/japplphysiol.00485.2020.

10. Caravaggi P, Pataky T, Goulermas JY, Savage R, Crompton R. A dynamic model of the windlass mechanism of the foot: evidence for early stance phase preloading of the plantar aponeurosis. J Exp Biol. 2009;212(15): 2491–2499. doi:10.1242/jeb.025767.

11. Caravaggi P, Pataky T, Günther M, Savage R, Crompton R. Dynamics of longitudinal arch support in relation to walking speed: contribution of the plantar aponeurosis. J. Anat. 2010;217(3): 254–261. doi:10.1111/j.1469-7580.2010.01261.x.

12. Pataky TC, Caravaggi P, Savage R, Parker D, Goulermas JY, Sellers WI, et al. New insights into the plantar pressure correlates of walking speed using pedobarographic statistical parametric mapping (pSPM). J. Biomech. 2008;41(9): 1987–1994. doi:10.1016/j.jbiomech.2008.03.034.

Comment: More detailed participant info, such as height, would be expected.

Response: Mean and standard deviation of participant height has been included, the text now reads as follows,

 “Fourteen volunteers (12M/2F; 22 ± 4 yrs; 64.6 ± 11.7 kg, 1.71 ± 0.08 m) participated in the study.”

Comment: If three trials were sufficient to reduce the data collection errors?

Response: Three trials (or less) have been used for a variety of recent foot-related biomechanical analyses (e.g., Bruening et al. 2012; Wager & Challis, 2016; Takahashi et al., 2017; Bruening & Takahashi, 2018; Deschamps et al., 2022) and we therefore are of the opinion that three trials sufficiently reduced errors in data collection. 

Bruening DA, Cooney KM, Buczek FL. Analysis of a kinetic multi-segment foot model. Part I: Model repeatability and kinematic validity. Gait Posture. 2012;35(4): 529–534. doi:10.1016/j.gaitpost.2011.10.363.

Takahashi KZ, Worster K, Bruening DA. Energy neutral: the human foot and ankle subsections combine to produce near zero net mechanical work during walking. Sci Rep. 2017;7(1): 1–9. doi:10.1038/s41598-017-15218-7

Wager JC, Challis JH. Elastic energy within the human plantar aponeurosis contributes to arch shortening during the push-off phase of running. J Biomech. 2016;49(5): 704–709. doi:10.1016/j.jbiomech.2016.02.023.

Bruening DA, Pohl MB, Takahashi KZ, Barrios JA. Midtarsal locking, the windlass mechanism, and running strike pattern: A kinematic and kinetic assessment. J Biomech. 2018;73(2018): 185–191. doi:10.1016/j.jbiomech.2018.04.010.

Deschamps K, Eerdekens M, Peters H, Matricali GA, Staes F. Multi-segment foot kinematics during running and its association with striking patterns. Sports Biomech. 2022;21(1): 71–84. doi:10.1080/14763141.2019.1645203

Comment: Has the author (s) considered the analysis of variance test to check the statistical inference between groups. 

Response: We indeed considered employing analysis of variance methods in the present study, but it would not be appropriate/necessary as there are only two levels for each would-be factor (Factor A: toe-wedge or no toe-wedge and Factor B: added mass or no added mass). With only two levels for a given factor, analysis of variance and t-tests are equivalent. 

Comment: Please note that t-tests are NOT the non-parametric tests.

Response: As the reviewer points out, the t-test is indeed not a non-parametric test. The probability density function itself was generated in a non-parametric manner. The text concerning the non-parametric t-test now reads as follows:

“If either test was statistically significant, paired t-tests were conducted by generating probability density functions non-parametrically using 10,000 unique permutations of the experimental data [31].”

Comment: I can not find the figure in the S1 appendix.

Response: Our apologies, the figure did not upload properly. S1 Fig has been included in the revision. 

Comment: What is PA stand for in the caption of Fig 3? The readers would appreciate the full name and abbreviation in the figure.

Response: The caption for figure 4 (originally Fig 3) now reads,

“Fig 4. Plantar aponeurosis (PA) Strain. Time-series profile of strain in the PA during stance for the added mass, control, and toe-wedge conditions. The shortest length at which there was tension in the PA for each participant was found using the mean of the shortest recorded PA lengths in each trial. The horizontal bars indicate timing of a statistically significant time-series differences between the toe-wedge and control conditions (top) and between the toe-wedge and added mass conditions (bottom) using two-tailed paired t-tests (α = 0.017 following Bonferroni correction).”

Comment: I recommend splitting Fig 5 with panels A and B for moment and quasi-stiffness measures.

Response: Fig 6 (originally figure 5) has been split into two panels per your recommendation.

Comment: It’s a bit hard to understand why the black line (AM) is wider than others. Presenting data with Mean±Standard deviation may help to illustrate the data trend.

Response: The black line is thicker only to aid in differentiating between lines if the figures are viewed in black-and-white and in situations when time-series values are very close to one another. We elected to illustrate the data trends by presenting the samples where there was a statistically significant difference in place of presenting Mean±Standard deviation to reduce the amount of additional information per figure. 

Comment: It is suggested that the figure legends be depicted below the subfigure Power and list in line.

Response: We have adjusted Figure 3 (originally figure 2) based on your recommendation. 

 

Responses to Comments of Reviewer 2

Comment: Thank you for the opportunity to review this interesting paper. This study sought to explore the role of the windlass mechanism and medial arch rigidity on energy transfer during walking by increasing dorsiflexion at the metatarsophalangeal joint (MTPJ) angle and increasing body mass. The work is an extension of work by Welte et al. (2018) who manipulated MTPJ angle during vertical loading and the work of Kern et al. (2019) who explored the effect of increased mass on midtarsal quasi-stiffness in walking. The study would appear to be novel and well designed and delivered. My comments are largely around adding greater explanation of rationale, mechanisms and variables for those who are not so familiar with potentially challenging concepts.

Response: Thank you for your positive comments and feedback.

Comment: Page 3, lines 58 to 61: Figure 1 in the paper by Welte et al. assists the explanation of the windlass mechanism well. The authors of the present study may wish to consider adding a similar figure.

Response: A figure detailing the influence of MTP joint dorsiflexion on the tension in the plantar aponeurosis has been included as Figure 1. The figure and caption are shown below:

Fig 1. Metatarsophalangeal joint dorsiflexion increases plantar aponeurosis tension. The windlass mechanism, wherein dorsiflexion of the metatarsophalangeal joint increases the tension in the plantar aponeurosis, potentially increasing the rigidity of the foot’s arch.

Comment: Similarly, clarity could be improved by defining key variables (e.g. quasi-stiffness) and establishing their relevance early on. For example, in the introduction in the Kern et al. paper it is stated: “Quasi-stiffness of the ankle (sometimes called dynamic stiffness) is defined as the slope of the joints’ moment-angle relationship (Sanchis-Sales et al., 2016; Shamaei, Sawicki & Dollar, 2013; Rouse et al., 2013). This is an experimentally derived parameter, which describes the joints’ resistance to motion for a given change in moment throughout stance.”

Response: The sentence introducing the study by Kern et al. (2019) has been edited to read,

 “Kern et al. [13] studied the rigidity of the midtarsal joint (a joint used to represent the MLA) when participants walked normally and with added mass, a perturbation which increases the forces and moments experienced by the foot’s joints and could thereby alter the function of structures which cross these joints. Rigidity was quantified using sagittal plane midtarsal joint quasi-stiffness, which is the slope of the resultant joint moment to joint angular excursion line.”

13. Kern AM, Papachatzis N, Patterson JM, Bruening DA, Takahashi KZ. Ankle and midtarsal joint quasi-stiffness during walking with added mass. PeerJ. 2019;7: e7487. doi:10.7717/peerj.7487.

Comment: Page 4, line 72: I would replace “In Kern et al” with “In a study by Kern et al” and “participant’s” with “participants’”.

Response: The sentence in question was adjusted to address the previous comment. 

Comment: Page 4, lines 72 to 80: The reporting of the findings of Kern et al is good, however I think the rationale for why increasing mass is worth investigating could be made clearer, perhaps with reference to altered forces experienced by the foot as the original authors did.

Response: This section has been edited to explicitly mention the rationale for using additional mass as a perturbation. It now reads,

 “Kern et al. [13] studied the rigidity of the midtarsal joint (i.e., arch) when participants walked normally and with added mass, a perturbation which increases the forces and moments experienced by the foot’s joints and could thereby alter the function of structures which cross these joints.”

13. Kern AM, Papachatzis N, Patterson JM, Bruening DA, Takahashi KZ. Ankle and midtarsal joint quasi-stiffness during walking with added mass. PeerJ. 2019;7: e7487. doi:10.7717/peerj.7487.

Comment: Page 5, line 98: Was there a justification for recruiting 14 participants? Was a power calculation performed?

Response: A power analysis based on the results of Welte et al. (2018) and Sichting et al. (2020) (who used insoles which curved upward beneath participant’s toes) indicated that large effect sizes (Cohen’s d of ~1) could be expected. Ten participants would be required to detect these effects at a power of 0.8. Some additional participants were recruited in case of data collection errors which rendered their data unusable. A variety of recent studies investigating foot mechanics have used between 9-14 participants (Welte et al., 2018; Takahashi et al., 2017; Kelly et al., 2018; Farris et al., 2019; Sichting et al., 2020).

Welte L, Kelly LA, Lichtwark GA, Rainbow MJ. Influence of the windlass mechanism on arch-spring mechanics during dynamic foot arch deformation. J R Soc Interface. 2018;15(145): 20180270. doi:10.1098/rsif.2018.0270.

Takahashi KZ, Worster K, Bruening DA. Energy neutral: the human foot and ankle subsections combine to produce near zero net mechanical work during walking. Sci Rep. 2017;7(1): 15404. doi:10.1038/s41598-017-15218-7.

Kelly LA, Cresswell AG, Farris DJ. The energetic behaviour of the human foot across a range of running speeds. Sci. Rep. 2018;8(1): 10576. doi:10.1038/s41598-018-28946-1.

Farris DJ, Kelly LA, Cresswell AG, Lichtwark GA. The functional importance of human foot muscles for bipedal locomotion. Proc Natl Acad Sci. 2019;116(5): 1645–1650. doi:10.1073/pnas.1812820116.

Sichting F, Holowka NB, Hansen OB, Lieberman DE. Effect of the upward curvature of toe springs on walking biomechanics in humans. Sci Rep. 2020;10(1): 14643. doi:10.1038/s41598-020-71247-9.

Comment: Page 5, line 103: The flow of the Methodology (Materials and Methods according to PLOS One submission guidelines) may be improved by reporting the protocol before the foot model, as was the case in the earlier work by Welte et al. and Kern et al.

Response: The experimental protocol has now been listed prior to the foot model.

Comment: Page 5, line 101: According to PLOS One submission guidelines it should be specified whether informed consent was written or oral.

Response: The word “oral” has been inserted here for clarity. 

Comment: Page 5, line 108: What was the diameter of the markers?

Response: Marker diameter was 12.7 mm. This has been included in the text which now reads,

“Bony landmarks for skin markers (diameter: 12.7 mm), joint centers, and segmental axes are depicted in Fig 2 and described in detail in S1 Appendix.”

Comment: Page 6, lines 118 to 126: For completeness I would add a specific definition of strain (length change relative to resting length?) in outlining how length of the PA was calculated. 

Response: The following text has been added for completeness,

 “PA strain was calculated as the change in length relative to this shortest length at which there was tension in the PA.”

Comment: I would also consider placing this section in the data analysis along with the definition of the kinetic variables of interest.

Response: The paragraph outlining the calculation of variables relating to the PA has been moved to the data analysis sub-section.

Comment: Page 6, lines 130 to 132: Were the markers attached directly to the skin and not on socks?

Response: The markers were attached directly to the skin. For clarity, the text now reads,

 “Bony landmarks for skin markers (diameter: 12.7 mm), joint centers, and segmental axes are depicted in Fig 2 and described in detail in S1 Appendix.”

Comment: Page 6, lines 128 to 129: The concept of a Froude number was interesting and not something I have come across before in gait or foot and ankle literature (and I see the reference is rather old). Is it known how this number differs from a self-selected speed? Could it be a point for discussion, seeing as Welte et al. compared loading at different speeds, although the only significant difference between speeds which was found as in energy dissipation? 

Response: The Froude number used in the present study resulted in target walking velocities which were similar to the self-selected speeds of young, healthy adults (Browning et al., 2006). Because Froude number is a function of leg length, participants that have longer legs walked slightly faster than those with shorter legs (target velocity ranged from 1.3 m/s to 1.47 m/s). While it is true that Welte et al. (2018) found differences in energy dissipation between loading rates, their ‘fast’ loading rate approximated the rates seen in walking whereas the ‘slow’ loading rate was less than half of the ‘fast’ loading rate. Therefore, the loading rate difference in their study is much larger than the difference expected between participants in the present study, even with some variation in walking speed between participants. Further, there was no statistically significant difference in walking speed between conditions in the present study, therefore it is unlikely that our energetic results are due to differences in walking speed. 

Browning RC, Baker EA, Herron JA, Kram R. Effects of obesity and sex on the energetic cost and preferred speed of walking. J Appl Physiol. 2006;100(2): 390–398. doi:10.1152/japplphysiol.00767.2005.

Welte L, Kelly LA, Lichtwark GA, Rainbow MJ. Influence of the windlass mechanism on arch-spring mechanics during dynamic foot arch deformation. J R Soc Interface. 2018;15(145): 20180270. doi:10.1098/rsif.2018.0270.

Comment: In any case I would recommend rearranging the sentence to start with something along the lines of “Target walking velocity was established by…” so the Froude number is not emphasized as much and does not detract from the main purpose.

Response: The text in question now reads,

 “Target walking velocity was established using a Froude number of 0.22 [14]. Briefly, the Froude number scales gait velocity based on participant leg length (measured from the floor to participant’s greater trochanter) and is used to produce ‘dynamically similar’ gait patterns for individuals of varying leg lengths [14].”

Comment: Page 6, line 133: Is the shore value (hardness) of the wedge known?

Response: The wedge was 3D printed using polylactic acid, which has a shore D value of between 83D (Ultimaker, 2017). The wedge was hard enough that it did not deform appreciably under load during gait. 

Technical data Sheet Pla - scan. Technical Data Sheet PLA. Ultimaker; 2017. Available: https://www.scan.co.uk/PDFs/Products/UltimakerPLA.pdf

Comment: Page 6, line 138: The previous work used 15% and 30%, why was a value of 15% chosen here?

Response: In the previous work, 15% added body mass was shown to increase the quasi-stiffness of the midtarsal joint, therefore it was deemed a large enough perturbation to employ in the current study. Further, there was no statistically significant pair-wise difference between the midtarsal joint quasi-stiffness of the midtarsal joint in the 15% and 30% body-weight added mass conditions in Kern et al. (2019). Lastly, Huang and Kuo (2014) found a linear relationship between some gait spatiotemporal variables and the magnitude of added mass in walking, therefore by using a lower added mass we were more confident that general gait dynamics would be similar between the added mass and control conditions. 

Kern AM, Papachatzis N, Patterson JM, Bruening DA, Takahashi KZ. Ankle and midtarsal joint quasi-stiffness during walking with added mass. PeerJ. 2019;7: e7487. doi:10.7717/peerj.7487.

Huang TP, Kuo AD. Mechanics and energetics of load carriage during human walking. J Exp Biol. 2014;217(4): 605–613. doi:10.1242/jeb.091587.

Comment: Page 7, line 152: Was there a rationale for a threshold of 35 N, which is higher than I would expect?

Response: The value of 35N was chosen as it was above the noise values in force plate signals for every participant. Indeed, for some participants a lower threshold could have been used, but a single threshold was preferred for consistency across participants.

Comment: Page 7, line 154: Is the rotation sequence correct? In reference 19 it is stated: “A ZYX Tait–Bryan angle sequence determined the angles of the first metatarsal relative to the calcaneus (arch angles) and the phalanx relative to the metatarsal (MTPJ angle)”.

Response: Welte et al. (2021) used different axes labels than in the current study, therefore their rotation sequence was YXZ when transferred into the axis definitions used here. ZXY was used in the present study instead of YXZ as it aligns with ISB recommendations for the ankle joint coordinate system (Wu et al., 2002) which proposed that the rotation sequence should be flexion / in-eversion / ab-adduction. As such, the reference to Welte et al. (2021) has been removed and replaced by a reference to Wu et al. (2002). 

Welte L, Kelly LA, Kessler SE, Lieberman DE, D’Andrea SE, Lichtwark GA, et al. The extensibility of the plantar fascia influences the windlass mechanism during human running. Proc R Soc B. 2021;288(1943): 20202095. doi:10.1098/rspb.2020.2095.

Wu G, Siegler S, Allard P, Kirtley C, Leardini A, Rosenbaum D, et al. ISB recommendation on definitions of joint coordinate system of various joints for the reporting of human joint motion—part I: ankle, hip, and spine. J Biomech. 2002;35(4): 543–548. doi:10.1016/S0021-9290(01)00222-6.

Comment: Page 8, lines 163 to 164: How was power calculated (vertical GRF multiplied by arch velocity?).

Response: The power due to deformation of structures distal to the rearfoot was calculated based on the methods in Siegel et al. (1996). Briefly, this power is calculated by taking the cross product of the free moment and the angular velocity of the rearfoot segment plus the cross product of the ground reaction force and the foot deformation velocity. Foot deformation velocity is approximated using the velocity of the rearfoot plus the linear velocity of the rearfoot relative to the center of pressure. The text has been edited to refer the reader to Siegel et al. (1996) for a detailed explanation,

 “MTP, midtarsal, and ankle joint six degrees of freedom power were calculated [23], as well as distal foot power which represents the power due to deformation of all structures distal to the estimated center of mass of the rearfoot segment (refer to [21] for distal foot power calculation details).” 

23. Zelik KE, Takahashi KZ, Sawicki GS. Six degree-of-freedom analysis of hip, knee, ankle and foot provides updated understanding of biomechanical work during human walking. J Exp Biol. 2015;218(6): 876–886. doi:10.1242/jeb.115451.

24. Siegel KL, Kepple TM, Caldwell GE. Improved agreement of foot segmental power and rate of energy change during gait: Inclusion of distal power terms and use of three-dimensional models. J. Biomech. 1996;29(6): 823–827. doi:10.1016/0021-9290(96)83336-7.

Comment: Page 9, line 196 and subsequent references: I do not see a supplementary figure, only tables and text…

Response: Our apologies, the figure did not upload properly. S1 Fig has been included in the revision.

Comment: Page 10 to page 13: When referring to differences shown in Figure 2, readability could be improved by referencing the specific panel in Figure 2.

Response: Panel lettering has been included for each panel in Figure 3 (originally figure 2) and reference to the specific panels has been included in the text.

Comment: Page 11, lines 232 to 240: My interpretation is that the circles and triangles represent individual participants, if this is correct it may help to clarify in the figure caption? I would advocate such an approach as there is a lot of inter-individual variability in foot function, so it is useful to demonstrate whether differences in conditions was consistent across participants in addition to any difference in the means.

Response: Each individual shape (circle or triangle) indeed corresponds to a given participant. The following text was included in the figure captions of Fig 5 and 6 for clarity,

“Values for each participant are displayed in each condition, with grey lines connecting a single participant across conditions.”

Comment: Page 16, lines 361 to 362: Was foot type/posture accounted for? How might this effect finings?

Response: Foot type was not accounted for in the present analysis. The study by Kern et al. (2019) examined arch height as an explanatory mechanism behind differences in midtarsal joint quasi-stiffness and found that it was unable to explain the variation. Further, Zifchock et al. (2006) found a weak (R2 = 0.09) relationship between arch height and arch stiffness (calculated using two discrete, static, measurements as opposed to in gait). Lastly, Holowka et al. (2021) examined arch height and midtarsal joint quasi-stiffness in running and found no statistically significant relationship. Based on the previous work investigating this question, we are of the opinion that foot type/posture does not explain the current results.

Kern AM, Papachatzis N, Patterson JM, Bruening DA, Takahashi KZ. Ankle and midtarsal joint quasi-stiffness during walking with added mass. PeerJ. 2019;7: e7487. doi:10.7717/peerj.7487.

Zifchock RA, Davis I, Hillstrom H, Song J. The Effect of Gender, Age, and Lateral Dominance on Arch Height and Arch Stiffness. Foot Ankle Int. 2006;27(5): 367–372. doi:10.1177/107110070602700509.

Holowka NB, Richards A, Sibson BE, Lieberman DE. The human foot functions like a spring of adjustable stiffness during running. J Exp Biol. 2020;224: jeb.219667. doi:10.1242/jeb.219667.

Comment: Page 16, lines 363 to 364: A “to” is missing from “[differences…] due active”

Response: The word “to” has been inserted accordingly.

Comment: Figures 2-5: Clarity may be improved by writing added mass and toe wedge in full rather than using uncommon abbreviations.

Response: Figures 3-6 (originally 2-5) have been edited accordingly.

---

## [Editor Report · Decision Letter 1]

23 Aug 2022

Foot arch rigidity in walking: *In vivo* evidence for the contribution of metatarsophalangeal joint dorsiflexion

PONE-D-22-15548R1

Dear Dr. Davis,

We’re pleased to inform you that your manuscript has been judged scientifically suitable for publication and will be formally accepted for publication once it meets all outstanding technical requirements.

Kind regards,

Imre Cikajlo, Ph.D.

Academic Editor

PLOS ONE
---

## [Editor Report · Acceptance letter]

30 Aug 2022

PONE-D-22-15548R1 

Foot arch rigidity in walking: *In vivo* evidence for the contribution of metatarsophalangeal joint dorsiflexion 

Dear Dr. Davis:

I'm pleased to inform you that your manuscript has been deemed suitable for publication in PLOS ONE. Congratulations! Your manuscript is now with our production department. 

Kind regards, 

on behalf of

Professor Imre Cikajlo 

Academic Editor

PLOS ONE